Marine diversity patterns in Australia are
filtered through biogeography. *Proc. R. Soc. B*
**288**: 20211534.

ecology

ecoregions, Fisher's alpha, latitudinal diversity,
mollusca, species-energy effect, provinciality

**Author for correspondence:**
Matthew R. Kerr
e-mail: matthew.kerr@cefas.co.uk

Electronic supplementary material is available
online at https://doi.org/10.6084/m9.figshare.
c.5680581.

# Marine diversity patterns in Australia are filtered through biogeography

Matthew R. Kerr[1] and John Alroy[2]

[1]Centre for Environment, Fisheries and Aquaculture Science, Lowestoft NR33 0HT, UK
[2]Department of Biological Sciences, Macquarie University, Sydney 2109, Australia

(iD) MRK, 0000-0002-0557-309X; JA, 0000-0002-9882-2111

Latitudinal diversity gradients are among the most striking patterns in
nature. Despite a large body of work investigating both geographic and
environmental drivers, biogeographical provinces have not been included
in statistical models of diversity patterns. Instead, spatial studies tend to
focus on species–area and local–regional relationships. Here, we investigate
correlates of a latitudinal diversity pattern in Australian coastal molluscs. We
use an online database of greater than 300 000 specimens and quantify diver-
sity using four methods to account for sampling variation. Additionally, we
present a biogeographic scheme using factor analysis that allows for both
gradients and sharp boundaries between clusters. The factors are defined
on the basis of species composition and are independent of diversity.
Regardless of the measure used, diversity is not directly explained by com-
binations of abiotic variables. Instead, transitions between regions better
explain the observed patterns. Biogeographic gradients can in turn be
explained by environmental variables, suggesting that environmental
controls on diversity may be indirect. Faunas within provinces are homo-
geneous regardless of environmental variability. Thus, transitions between
provinces explain most of the variation in diversity because small-scale fac-
tors are dampened. This explanation contrasts with the species-energy
hypothesis. Future work should more carefully consider biogeographic
gradients when investigating diversity patterns.

## 1. Introduction

Global patterns of species diversity are a central focus of ecological and biogeo-
graphic research. In particular, there has been substantial discussion of declines
in diversity from tropical to temperate latitudes, a pattern seen in most major
groups [1,2] and in the fossil record [3–6]. Many explanations for this pattern
focus on evolutionary drivers, with higher rates of speciation in the tropics
[4,7,8]. However, higher rates of speciation at the poles [9] and marginal effects
of latitude [10] have also been reported.

Latitudinal effects have also been found to control species richness in a
number of taxa. At continental scales, however, biogeographic boundaries do
affect species richness patterns. For example, in North America, richness pat-
terns of bivalves are stepwise and match provincial boundaries [11].
Additionally, biogeographic factors have been found to be a primary predictor
of richness in tropical fish [12,13]. Biogeographic boundaries have also acted as
a control on changes in diversity and species traits on long timescales [14].
A possible explanation is that within provinces, diversity is constant because
local faunal assemblages are homogeneous, overwriting fine-scale environ-
mental signals. This would explain why transitions between provinces drive
larger diversity patterns. That said, biogeographic boundaries themselves reflect
shifts in environmental regimes: for example, they can correspond with moving
from a sub-tropical to temperate realm [11]. Thus, geographic structures of
marine benthic faunas have in turn been linked strongly to environmental
variables [15].

Despite these references to biogeographic controls, most recent studies do not include provinces in models of global biodiversity [16–20], despite the usefulness of province definitions for tracking species distributions and delineating protected areas [21]. Instead, most references to geographic structure in species communities are focused on local-regional diversity patterns (e.g. [22–25]) and are not directly concerned with controls on latitudinal gradients. Additionally, historical biogeographic effects on richness have been reported, but they have been interpreted as a function of productivity and abiotic environmental variables [26]. Gradients in modern biogeographic structure have largely been overlooked.

Indeed, the majority of the literature focuses only on environmental correlates, linking diversity to global variation in climate and productivity [27–29]. Of particular interest is the consistent link between diversity and temperature [30], often used as a proxy for solar energy. This is suggested to support higher diversification rates through fostering higher metabolic rates and therefore mutation rates [7,28–30]. Climate seems to control fluctuations in species richness through time, both in simulation [31] and in empirical data [32]. Specifically, in the marine realm, temperature has been found to be the primary control on richness for a variety of marine groups [16,17,19,33–36], with nutrient levels also acting as a correlate for macroinvertebrates [16] and benthic species in general [34]. Depth also influences global marine biodiversity in the open ocean [20,28,37].

Here, we test the hypothesis that transitions between biogeographic regions more directly govern molluscan richness in coastal Australia than does spatial variation in abiotic conditions. We ask whether homogeneity of faunal composition overwrites the environmental signal within provinces, leaving biogeographic transitions as the main driving factor of continental-scale diversity patterns. We note that Australia is included in many global analyses, but rarely features in continental-scale biodiversity studies. We define provinciality using multivariate methods, not only to illustrate relationships but to showcase how diversity tracks province boundaries. Using this approach, we demonstrate that we are working with indistinct boundaries. Using the new province scheme for Australia identified, we then investigate whether biogeography more directly controls diversity patterns than abiotic variables.

## 2. Material and methods

### (a) Data

All molluscan records were downloaded from the Ocean Biodiversity Information System (OBIS) on 15 January 2020 using the *robis* package in R [38]. Nomenclatural validity was checked by comparing taxa names to the World Register of Marine Species using the R package *worms*, ensuring that any non-marine species were excluded. Any record without a full species name was omitted, along with any record that did not have a latitude or longitude value resolved to at least two decimal places. Data were pooled into 0.5° cells to match the spatial resolution of the abiotic data (see below), and any cell containing fewer than 10 species or fewer than 150 records (see Diversity Estimation) was excluded from the analysis. The final dataset drawn from OBIS numbered 300 210 specimens of 6323 species located in 233 cells.

As OBIS data only record presence and absence, which is only a proxy for absolute abundance, we used two datasets from the Reef Life Survey (RLS: https://reeflifesurvey.com/) to test our results. RLS is a global citizen science project, and it uses data collected by trained divers who survey transects in reef systems [39]. RLS has a particular concentration of data from Australia. We downloaded the full dataset for fish and mobile macroinvertebrates and calculated richness and diversity in cells using the same methods as with the OBIS dataset. Results based on these datasets mirror those based on the OBIS dataset: all the resulting maps and regression coefficients are in the electronic supplementary material, with some comments on the results included in the main body of text. Because the RLS dataset focuses on reef environments and our goal is to examine the entire coastline of Australia, we focus here on the OBIS dataset.

Eight abiotic variables for the coastline of Australia were downloaded from the CSIRO Atlas of Regional Seas ([40]: www.cmar.csiro.au/cars). These are mean annual sea surface temperature, yearly temperature standard deviation, mean annual salinity, yearly salinity standard deviation, mean annual dissolved oxygen content and mean annual nitrate, silicate and phosphate content. The data consist of ocean properties, gridded on a 0.5° scale, generated from a variety of data sources. To compute annual values, we took the overall means of the latest daily environmental layers. As the majority of our data are coastal records, and due to the large grid scale, we did not include depth in our analyses. We also did not include coral reef area or shelf-area, for similar reasons.

To eliminate collinearity, abiotic variables were translated into loadings by varimax factor analysis [41,42] using the function *fa* in the R package *psych* [43]. We prefer a multivariate method over a model selection approach as it preserves information in the form of factor loadings, instead of potentially excluding variables that are of biological interest. A three-factor solution was favoured based on parallel analysis and inspection of a scree plot (electronic supplementary material, figure S1). Loading values for each abiotic factor are shown in the electronic supplementary material, table S1.

### (b) Diversity estimation

Despite good overall geographical coverage of coastal Australia in this dataset, there is substantial variation in sample size between degree cells (electronic supplementary material, figure S2). To account for the spatial bias inherent in using raw species counts as an estimate for species richness, we used a series of diversity and richness estimators. We assessed sample size by considering the number of presences for each species in each cell.

To guarantee that the results were robust, we used four different measures of diversity. First, we used species richness extrapolation. The Chao 1 estimator [44] is widely used for this purpose. It yields similar results to the corrected jackknife (cJ1) equation of [45]. However, cJ1 is more stable (electronic supplementary material), so we emphasize it in the main body of text. Second, we applied Fisher's $\alpha$ [46], which is typically used in studies of highly diverse systems [46,47] and generally stable at increasing sample sizes. Third, we used Simpson's D—a common diversity estimator with no sample size influence [48–50]. Finally, we used the analytical version [51] of shareholder quorum subsampling (SQS—[52–54]) to further account for sampling issues. We note that SQS is routinely referred to as coverage-based rarefaction (CBR) in the ecological literature, but not the palaeontological literature, and that the distinction between SQS and CBR is not conceptual but operational, as with the distinction between the original formulation of rarefaction [55] and the analytical formulation [48] that is now widely used.

Each of these methods reduces the effect of sample size on diversity estimates (electronic supplementary material, figure

S3) relative to using simple counts of species, which are common in large-scale studies of diversity. Estimates do increase with sample size until around 150 individuals, even for Fisher's $\alpha$ (electronic supplementary material, figure S3), so any cells with fewer than 150 individuals were removed from the analyses. To ensure consistency with previous studies using similar data, we also used face-value counts of species that are uncorrected for sampling variation. Analyses based on these counts yielded the same model results as with the corrected data, but they showed no latitudinal pattern. They are detailed in electronic supplementary material, figure S4 and table S2.

## (c) Biogeographic assignment

Instead of using the constantly changing qualitative schemes of Australian marine biogeography [56–59] objective bioregions were defined from the ecological data using two approaches. First, clusters were identified using partitioning around medoids (PAM) clustering [60]. PAM clustering generates more consistent clusters than $k$-means due to minimizing the influence of outliers [61,62].

Second, loadings generated from a varimax factor analysis were generated—cells which score highly on a similar factor can be considered to represent the same ecological space, with middling scores representing transition zones. Loadings generated are more intuitive and, unlike PAM, test for both sharp ecological boundaries (such as those assumed in previous biogeographic studies (e.g. [11,12])) and smooth gradients. To guarantee cluster integrity, a correspondence analysis [63] was applied to the dataset and used to demonstrate the distinctiveness of clusters.

Optimal factor counts were likewise generated for each dataset using parallel analysis, which compares the scree patterns of factor eigenvalues with screes generated from random data matrices of the same size [64]. These values were favoured over manual inspection of a scree plot as they are not subjective, although we include scree plots in the electronic supplementary material for comparison. We implemented both factor analysis and parallel analysis using the R package *psych* [43].

## (d) Analysis

Spatial autoregressions were carried out to compare the generated richness values to both the abiotic variables and the abiotic factor scores, allowing for spatial autocorrelation in the abiotic scores using the R package *spatialreg* [65]. An additional model was run that included the biogeographic loadings. Using factor scores as variables allowed provinciality to be included intuitively in the model because the resulting slopes reflect the strength of provincial signals. This analysis was then repeated with each richness metric, and for all cluster counts supported by the scree plot and parallel analysis. In addition to spatial autoregressions, we carried out linear multiple regressions that are numerically almost identical to the spatial autoregressions. Results are included in the electronic supplementary material.

To further test the relationships between abiotic variables, biogeographic loadings and diversity, we ran several additional multiple regressions that emulated a structural equation model. The key abiotic variables were chosen by identifying the variable with the highest loading on each of the three abiotic factor scores. Abiotic factor scores with the highest uniqueness values were also chosen because a high uniqueness indicates that the variance of each variable was not explained by the factors. This narrowed the set of key variables to seven: latitude, mean annual sea surface temperature, yearly temperature standard deviation, mean annual salinity, yearly salinity standard deviation, mean annual dissolved oxygen content, and mean annual silicate content. Each dependent variable was then run in a multiple regression

framework against each other variable in turn. Any significant relationship between two variables was recorded. Factor scores representing biological provinces were not run against each other as they are mathematically related—a high score in one province will generate a low score in other provinces.

To avoid circularity in the assignment of biogeographic provinces based purely on species distribution, a simulation was run that deliberately removed geographical and habitat distribution and focused on temperature. A temperature range for each species was generated from the data using the minimum and maximum annual temperature for each cell the species occupied. A presence–absence matrix was then randomly generated, filling each cell with species that overlapped in their temperature ranges and then randomly sampled so that it contained the same number of species per cell as the original matrix. This created a simulated presence–absence matrix, where species presences are only controlled by temperature and not directly by geography. A factor analysis generated nominal biogeographical provinces for the sampled matrix, using the same number of provinces found in the real data. A regression model identical to the one used in the original analysis was then used to assess the relationship between simulated biogeography, temperature and richness. As this process is stochastic, we repeated the subsampling process 10 000 times and recorded the variation in model results across replications.

# 3. Results

## (a) Species richness and biogeography

Australian molluscs exhibit a flattened diversity gradient, with high diversity in the tropics and a shallow decline southward (figure 1). Simpson's D and SQS indicate a slightly steeper decline in diversity along with the southern coastline (figure 1). Overall, there was agreement across methods used on the overall spatial diversity pattern.

Parallel analysis suggests there are five factors for the OBIS dataset, six for the RLS fish dataset, and three for the RLS invertebrate dataset. Scree plots of the OBIS data suggest three factors are appropriate, so we have included this number in electronic supplementary material, analyses.

These provinces are distinct from each other geographically (figure 2) and in ordination space (electronic supplementary material, figure S8). Both the OBIS and RLS datasets show a broad northern and southern province spanning the western to eastern coastlines, with smaller subdivisions present along with the eastern and southeastern coastlines seen when higher cluster counts are considered. Factor analysis shows that although cluster cores are distinct, there are broad transition zones on the southern and western coastlines as provinces become less distinct (figure 2; electronic supplementary material, figure S9). The Great Barrier Reef, northern Australia and eastern Australia provinces have much sharper boundaries (figure 2; electronic supplementary material, figure S9). In the northeast, coastal and offshore reefal cells run in parallel over a large latitudinal extent when five provinces are visualized across the region (figure 2b).

## (b) Regression and simulation analysis

When all abiotic variables were tested in one model, only annual temperature variation significantly explained the changes seen in species diversity for any dataset ($p < 0.001$). When combined into abiotic factors, factor 1, with strong

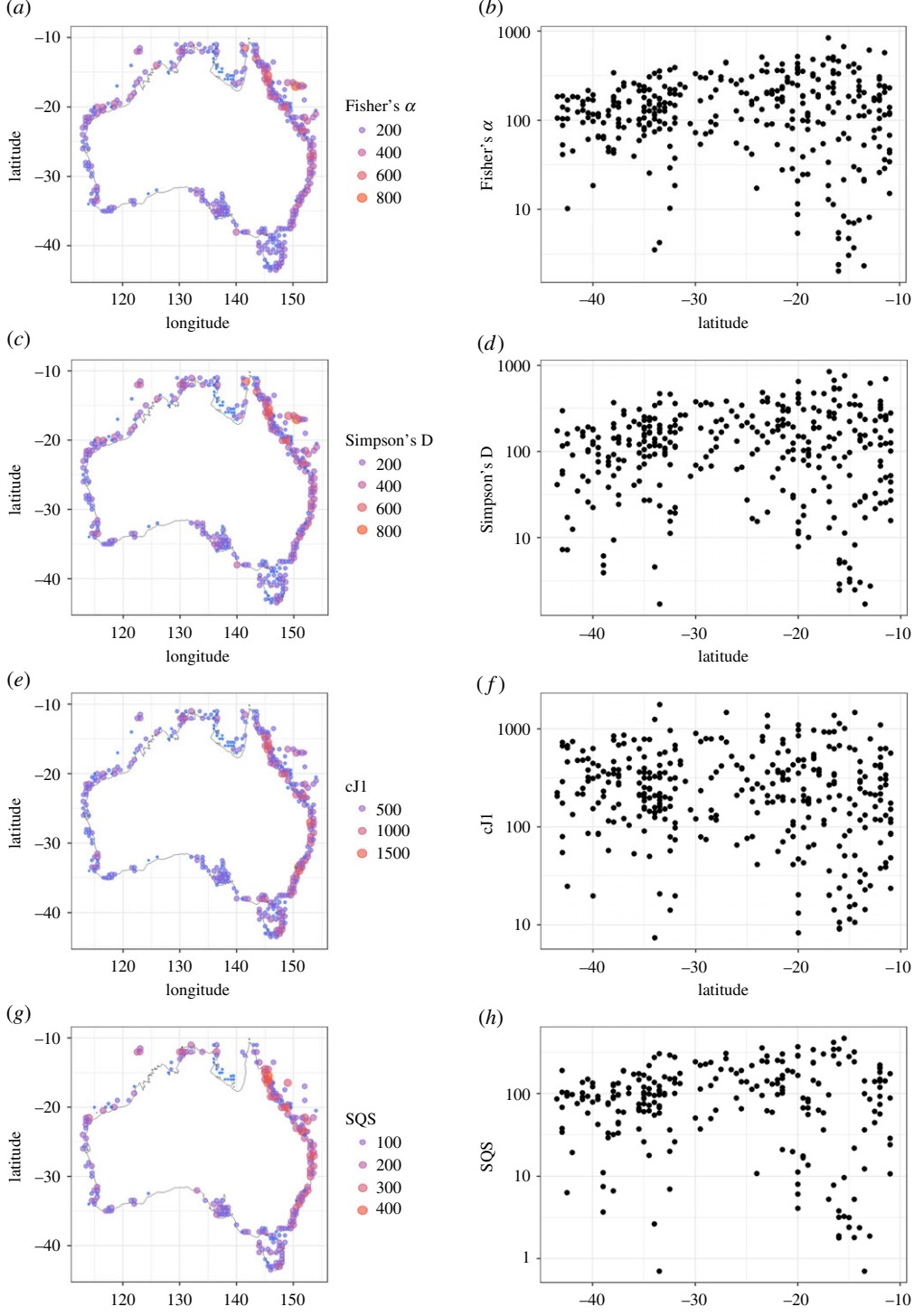

**Figure 1.** Spatial diversity patterns for Australian marine molluscs from the OBIS dataset, pooled into 0.5° cells. Four different methods are shown: Fisher's α (a,b), Simpson's D (c,d), the corrected first-order jackknife (cJ1—e and f), and shareholder quorum subsampling, also known as CBR (SQS—g and h). Point size and colour are scaled with diversity in (a), (c), (e) and (g). The y-axis in each scatter plot is on a log scale for clarity. (Online version in colour.)

loadings of temperature variables and dissolved oxygen content, was the only factor that could explain the richness pattern (table 1), and only for estimates generated using the corrected jackknife equation. Variance explained by each model was low, between 9% (cJ1) and 15.7% (Simpson's D).

When the factor loadings for the biogeographic provinces were added as explanatory variables, the abiotic variables were no longer significant predictors of diversity patterns. Instead, factor loadings relating to the eastern, northern and Great Barrier Reef provinces were significantly and positively related to diversity ($p < 0.01$: table 2; electronic supplementary material, table S3). The combined analysis explained up to 46% of variation across the dataset. These results are comparable to those generated from the RLS for fish (electronic supplementary material, table S4). The analysis of the RLS marine invertebrates (electronic supplementary material, table S5) suggested that the same variables are significant predictors, but they explained a much lower amount of variance.

When biogeographic factors were compared to abiotic factors in several multiple regressions, strong relationships were found for each province—but the important abiotic variables associated differed (figure 3). Temperature and dissolved oxygen content

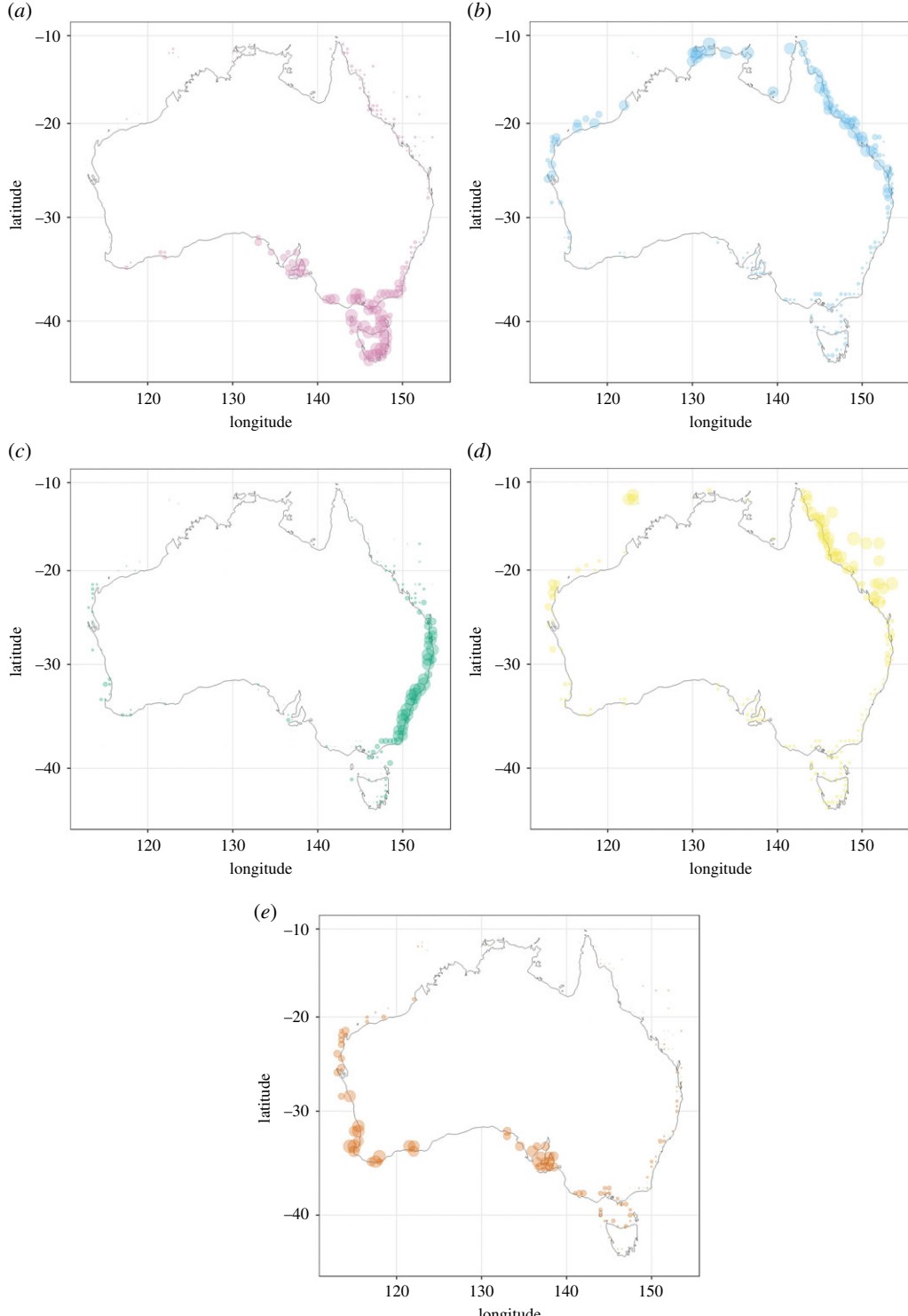

**Figure 2.** Biogeographic factors for coastal Australia, based on molluscan data downloaded from OBIS. Each panel displays a different factor deriving from a five-factor solution. Points are scaled so that factor loadings are reflected by their size. Provinces are as follows: (a) southeastern Australia, (b) northern Australia, (c) eastern Australia, (d) Great Barrier Reef and (e) southwestern Australia. For a map containing all provinces overlayed, see electronic supplementary material, figure S9. (Online version in colour.)

were significant for the western province and the southeastern province, temperature and silica content were significant for the northern Australia province, salinity variation was significant for the southwestern Australia province, and mean annual salinity was significant for the southeastern Australia province. No environmental variable was significantly associated with the Great Barrier Reef province.

When species ranges were simulated based on temperature rather than geographic position, biogeographic provinces formed east–west bands. The temperature became a significant predictor across trials and biogeography remained significant as in the initial analyses (electronic supplementary material, figure S11).

## 4. Discussion

The latitudinal diversity pattern in Australia is strikingly different to that observed across other continents and at the global scale [11,36,66]. Rather than the stepwise declining

**Table 1.** Results of a series of general linear model analyses of marine mollusc diversity against abiotic variables. Abiotic variables were collapsed into factors to avoid collinearity (for details of each factor see electronic supplementary material, table S1) and scaled so they are beta coefficients and therefore comparable. Values shown are slope estimates generated from the model; italicized values are significant ($p < 0.001$).

| diversity estimate | abiotic factor one | abiotic factor two | abiotic factor three | adjusted $R^2$ |
|---|---|---|---|---|
| Fisher's $\alpha$ | 0.0824 | 0.0462 | 0.0415 | 0.008 |
| Simpson's D | 0.1204 | −0.0291 | 0.0013 | 0.000 |
| corrected jackknife | *0.1703* | 0.0640 | 0.0597 | 0.068 |
| SQS | 0.1310 | −0.0272 | 0.0120 | 0.0069 |

**Table 2.** Results of a series of general linear model regressions of marine mollusc diversity against biogeographic factors and abiotic variables. Abiotic variables were collapsed into factors to avoid collinearity (for details of each factor see electronic supplementary material, table S1) and scaled so they are comparable. A five-factor biogeographic solution is shown; for a three-factor solution, see electronic supplementary material, table S3. Provinces are as follows: 1 = southeastern Australia, 2 = northern Australia, 3 = eastern Australia, 4 = Great Barrier Reef, 5 = southwestern Australia (figure 2). Values shown are slope estimates generated from the model; values in italics are significant ($p < 0.001$).

| diversity estimate | province one | province two | province three | province four | province five | abiotic factor one | abiotic factor two | abiotic factor three | adjusted $R^2$ |
|---|---|---|---|---|---|---|---|---|---|
| Fisher's $\alpha$ | 0.106 | *1.538* | *1.661* | *1.979* | 0.498 | 0.068 | −0.023 | 0.049 | 0.440 |
| Simpson's D | −0.370 | *1.859* | *1.823* | *2.281* | 0.901 | 0.125 | −0.100 | 0.021 | 0.317 |
| corrected jackknife | *1.183* | *2.412* | *2.101* | *2.266* | 0.782 | 0.116 | −0.028 | 0.045 | 0.460 |
| SQS | 0.174 | *1.914* | *1.862* | *2.218* | 0.819 | 0.112 | −0.048 | 0.013 | 0.366 |

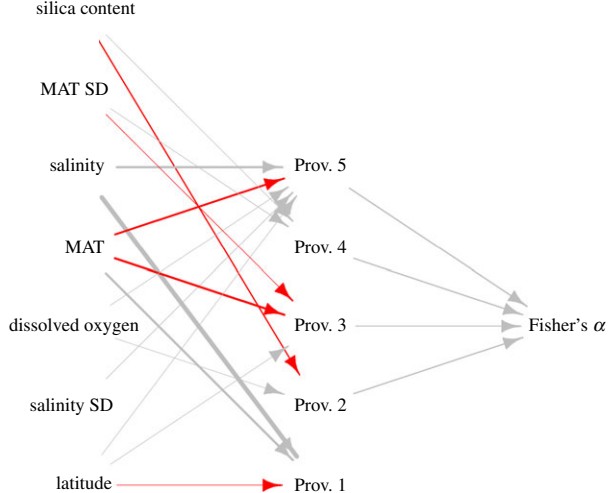

**Figure 3.** Relationships between biogeographic provinces, diversity (based on Fisher's $\alpha$) and environmental variables. Relationships were derived from a series of multiple regressions; a red arrow shows a significant negative relationship ($p < 0.01$) and a grey arrow shows a significant positive relationship. MAT = mean annual sea surface temperature, SD = annual standard deviation, and provinces are as follows: 1 = southeastern Australia, 2 = northern Australia, 3 = eastern Australia, 4 = Great Barrier Reef, 5 = southwestern Australia.

pattern observed in similar taxa along with eastern and western North America [11], we observed a gradual latitudinal decline. Species diversity differed between the Australian eastern and western coastlines, with much lower diversity along with the western coastline. In contrast with global studies, the gradient in this decline is very shallow, with smaller differences between tropical and temperate areas. This flattened gradient is apparent in models of other taxa that include regional diversity and report relative richness (e.g. [28]) but is vastly different to gradients observed in global analyses [66]. The abiotic variables we used could explain diversity patterns, but they explained a much lower amount of variation than observed in other studies; for example, Tittensor *et al.* [36] obtained a pseudo-$R^2$ of 70 to 90 for multiple groups using global data.

The provinces we present here are generally similar to existing schemes for marine Australia [57], with the Great Barrier Reef as a separate province to coastal Australia, a broad northern province and several smaller provinces in the southeast. At a broader scale, the data support the north–south divide observed in both global regionalization studies [67] and local molluscan provincial definitions [68]. Our factor analysis approach reveals the relationship between the provinces in greater detail with province boundaries in coastal Australia reflected by transition zones that cover a few degrees of latitude and are smaller than the zones previously illustrated in continental maps [56]. When only two factors are used, transition zones are both smaller and farther north than suggested by local studies on molluscs [68]. Studies on eastern Australian corals also point to a broad transition zone around the same latitude [69]. Cluster analysis reveals similar province distributions to the factor analysis. However, the clusters are rarely geographically consistent and fail to define sharp boundaries at this scale accurately, as they are meant to do. This effect is easily seen when three clusters are used: cluster analysis cannot resolve the southern coastline as belonging to either province.

Although diversity has been related to biogeographic boundaries before [11–13] and biogeographic boundaries have been related to environmental variables [13], there have been few studies resolving the relationships among the three in an intuitive way. Defining provinces with a factor analysis allowed provinces to be included in the models used here in an intuitive way. Province loadings were found to be the only significant explanatory variable, with no environmental variable significantly explaining richness. We show that there is a unidirectional causal pathway from environment to biogeography to diversity—although the relationships are inconsistent and different environmental variables correlate with different biogeographic factors. Therefore, province boundaries reflect abiotic conditions and richness in turn is controlled by major province boundaries. Diversity is instead largely homogeneous within factors, regardless of environmental variability, and the average environmental differences between provinces driver large-scale diversity changes. This is expected, as molluscan distribution is controlled by environment and current regimes [70–72] and abiotic stressors at range-edges [73] that could be reflected in biogeographic transitions. The mechanism may be that the same species pool is sampled throughout each province, resulting in uniform richness values. Note that biogeographic regions remain significant predictors of diversity even under a simulation that removes east–west biogeography.

Using two completely different datasets and groups of organisms—reef fish and macroinvertebrates—the same provinces were observed. As biogeographic factors explain twice as much variation in diversity as environmental factors, both environmental ranges of individual species and historical effects likely maintain the diversity pattern we see here.

Although we did not include depth and shelf-area, these are important covariates in prior studies of marine richness patterns [12,18,25,29]. It would be interesting to see higher resolution future studies include them in similar analyses to further test the relationships we uncover.

In sum, we show here that temperature, often cited as a major control on species diversity gradients, does not directly control species diversity at a continental scale and instead has an indirect influence through the control of provinces. This result highlights the need for diversity studies to more often include biogeographical provinces in their models, as the relationship between the environment and diversity may be only indirect.

Data accessibility. Code for downloading and processing occurrence data, extracting abiotic data from the CARS dataset and relevant richness functions have been included in the electronic supplementary material [74].

Authors' contributions. M.R.K.: conceptualization, data curation, formal analysis, funding acquisition, investigation, methodology, project administration, visualization, writing-original draft, writing-review and editing; J.A.: conceptualization, formal analysis, funding acquisition, methodology, project administration, supervision, validation, writing-review and editing. All authors gave final approval for publication and agreed to be held accountable for the work performed therein.

Competing interests. We declare we have no competing interests.

Funding. This study was funded by an international Macquarie University Research Excellence Scholarship (iMQRES; no. 2016343 to M.R.K.).

Acknowledgements. The authors would like to thank Mark Costello, James Crampton, John Pandolfi, Hudson Pinheiro, an anonymous reviewer, and the Palaeobiology Laboratory at Macquarie University for comments that improved this manuscript.

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
