## [Peer Review File · Proceedings of the Royal Society B: Biological Sciences]

Review History

RSPB-2021-1534.R0 (Original submission)

Review form: Reviewer 1

Recommendation

Accept with minor revision (please list in comments)

Scientific importance: Is the manuscript an original and important contribution to its field?

Good

General interest: Is the paper of sufficient general interest?

Good

Quality of the paper: Is the overall quality of the paper suitable?

Good

Is the length of the paper justified?

Yes

Should the paper be seen by a specialist statistical reviewer?

No

Do you have any concerns about statistical analyses in this paper? If so, please specify them explicitly in your report.

No

It is a condition of publication that authors make their supporting data, code and materials available - either as supplementary material or hosted in an external repository. Please rate, if applicable, the supporting data on the following criteria.

Is it accessible?

Yes

Is it clear?

Yes

Is it adequate?

Yes

Do you have any ethical concerns with this paper?

No

Comments to the Author

The authors have provided a robust analytical approach to delineate biogeographical provinces of Australian marine molluscs and validated the results using an independent data set. I enjoyed reading this manuscript and appreciate the approach taken, being using ecological data to identify bioregions. I believe this manuscript is suitable for publication, after some clarifications have been made and some minor issues addressed.

The major aspect which I would like clarified is the potential for bioregions/provinces used in the analyses to be acting as proxies for unmeasured environmental and/or geographical variables/features. Within the text the authors have mentioned "Biogeographic factors can in turn be explained by environmental variables, suggesting that environmental controls on diversity may be indirect" (line 17), in this instance does "indirect" refer to context dependent controls, or that environmental controls are moderated by other processes? I think that the relevance of the study would be greatly improved if the authors could clarify the statements such as this through the manuscript (e.g. line 269), as at times these statements appear to be circular. I did appreciate the insight provided on line 270, with important environmental variables differing among bioregions, I believe expanding on this would clarify the author's interpretation of the results. Similarly, the importance of scale of observations could also be mentioned, i.e. if 0.25° cells were used as opposed to 0.5° cells, would the relative importance of environment vs bioregional provinces be likely to differ? I'm not asking the author's to re-analyse the data at varying scales, but it might be worth noting in the discussion.

Minor comments

- Line 7, consider replacing "Has" with "have"
- Line 31, consider moving the comma to after "however"
- Line 33-35, consider describing a couple of examples in the text
- Line 70, - provide RLS internet address (e.g. line 80)
- Line 71, consider replacing "divers in training" with "trained divers"
- Line 80, are there any reasons depth and chlorophyll were not included to represent the environment?
- Line 83, what was the grid cell size/resolution?
- Line 125, would 'distinctiveness' (or similar) be better than 'visibility'?
- Line 129, consider replacing "implement" with "implemented"
- Line 144, maybe list the six abiotic variables here or in results
- Line 153, consider changing "then sampled" to "then randomly sampled" for clarification

- Line 192, is this supposed to be $p < 0.001$?
- Line 194-210, check table references (i.e. Table 3.1 etc)
- Line 235, possibly expand a little on the difference in the nature of diversity between east and west coasts
- Line 235, Please briefly expand on comparison to global studies, does Australia have depauperate tropical fauna or rich temperate fauna by global standards?
- Line 237, consider replacing "some" with "other"
- Line 245, maybe reword, line 182 refers to sharp boundaries on the east coast
- Line 264, consider replacing 'allows' with 'allowed'
- Line 270, I do like this insight
- Line 274-276, Please clarify
- Line 279, reference 15 only considered fish, not invertebrates

Figure "S3.3" caption needs correcting and supplementary numbers need fixing

Review form: Reviewer 2 (Hudson Pinheiro)

Recommendation

Accept with minor revision (please list in comments)

Scientific importance: Is the manuscript an original and important contribution to its field?

Good

General interest: Is the paper of sufficient general interest?

Good

Quality of the paper: Is the overall quality of the paper suitable?

Good

Is the length of the paper justified?

Yes

Should the paper be seen by a specialist statistical reviewer?

Yes

Do you have any concerns about statistical analyses in this paper? If so, please specify them explicitly in your report.

Yes

It is a condition of publication that authors make their supporting data, code and materials available - either as supplementary material or hosted in an external repository. Please rate, if applicable, the supporting data on the following criteria.

Is it accessible?

Yes

Is it clear?

Yes

Is it adequate?

N/A

Do you have any ethical concerns with this paper?

No

Comments to the Author

Dear Dr. Kerr and Dr. Alroy,

I enjoyed reading the paper and believe that the results are very interesting and have a lot to contribute to the field of biogeography. I have very few suggestions and questions. My main concern is related to the Discussion section, which could be improved exploring why distinct biogeographic provinces could better predict species richness than abiotic variables. Please find below my specific comments,

Sincerely,

Hudson Pinheiro

(I waive anonymity)

Marine diversity patterns in Australia are filtered through biogeography

Abstract

Line 11 – What do you mean by biogeographical controls?

Line 16 – it is not clear what are the biogeographic factors – species richness? Species composition? And what about the sharp boundaries? Did you only find evidences for the gradient?

Line 17 – These factors need to be clarified because the richness gradient is a biogeography pattern. The gradient is not related to environmental data but the biogeographic factors are, which in turn explain the gradient.

Line 19 – could you try to add a hypothesis for the direct effects of the biogeographic gradients on the richness patterns. The way it is now the conclusion is that biogeographic gradients explain richness patterns. It is not very clear what you meant because a richness pattern could be interpreted as the same as a biogeographic gradient.

Introduction

Line 24 – and even more important to the field of Biogeography.

Line 31 “biogeographic boundaries affect species richness patterns” – why? How? Do you think it is related to biogeographic barriers? These boundaries mark changes in endemic species, they usually are related to biogeographic barriers or changes in environmental conditions (e.g. transition of tropical to subtropical waters). I suggest you to develop a bit more about this subject.

Line 33 – “temporal changes” – you mean in an ecological or historical scale?

Line 36-38 – some studies on coral reef fishes have included regions and provinces in the models, accounting to the different evolutionary histories of them – See Parravicini et al 2013 (Ecography doi: 10.1111/j.1600-0587.2013.00291.x), Quimbayo et al 2019 (Ecography doi: 10.1111/ecog.03506).

Line 52-53 – It still not clear what do you mean as biogeographic gradient driving richness patterns because both could be used as synonyms depending on the context. You should describe more straightforward what you mean by biogeographic gradient and how it differ from richness gradient in the Introduction.

General remark - what about the diversity gradients explained by the hypotheses of Center of origin, accumulation, overlap? Mora et al 2003 (10.1038/nature01421.1.) found evidence for the effects of dispersal driving the gradient. Many other studies have found coral reef area as important driver for the diversity gradient as well.

Methods

Line 79 – What about shelf area or coral reef area, which is one of the main factors explaining the diversity of reef organisms, and it is such an important habitat in Australia? Did you try adding them into the model? And what about depth?

Line 155 – “species presences are only controlled by temperature and not directly by geography”
– How did you control for the cases where species are restricted to shore or offshore environments? Estuarine dependents or highly mobile species that approach offshore reefs? What if these different geographically habitats have same temperature?

General remarks: what about the description of the general linear models of marine mollusc diversity against abiotic variables?

Results

Figures – All the panels of the Fig 1 are very similar, so I suggest you to leave only one and send the others to supplementary material. In Figure 2, did you try to create a single figure with all the factors together? I think it would work fine and it will reduce the space for the journal.

Line 190 – suggest improving the heading, it is not informative.

Discussion

What I understood is that biogeographic factors is the biogeography of marine organisms in Australia, i.e., subprovinces based on the composition of the species. If the biogeography is more important to predict the richness in the region, the authors should discuss about that. If species composition is more important than the environment, the authors should discuss the great body of work that explore species interactions as determinant to the expected richness found in a province. Seminal papers by Ricklefs are very important to be mentioned. Others by Mora and Bellwood also explore the effects of dispersal and coral species richness and area on the fish richness. The authors could mention the existence of biogeographic barriers as filters, and the role of endemism in determining biogeographic and richness patterns.

Decision letter (RSPB-2021-1534.R0)

17-Sep-2021

Dear Mr Kerr:

Your manuscript has now been peer reviewed and the reviews have been assessed by an Associate Editor. The reviewers' comments (not including confidential comments to the Editor) and the comments from the Associate Editor are included at the end of this email for your reference. As you will see, the reviewers and the Editors have raised some concerns with your manuscript and we would like to invite you to revise your manuscript to address them.

When submitting your revision please upload a file under "Response to Referees" - in the "File Upload" section. This should document, point by point, how you have responded to the reviewers' and Editors' comments, and the adjustments you have made to the manuscript. We

require a copy of the manuscript with revisions made since the previous version marked as 'tracked changes' to be included in the 'response to referees' document.

Research ethics:

Use of animals and field studies:

It is a condition of publication that you make available the data and research materials supporting the results in the article. Please see our Data Sharing Policies (<https://royalsociety.org/journals/authors/author-guidelines/#data>). Datasets should be deposited in an appropriate publicly available repository and details of the associated accession number, link or DOI to the datasets must be included in the Data Accessibility section of the article (<https://royalsociety.org/journals/ethics-policies/data-sharing-mining/>). Reference(s) to datasets should also be included in the reference list of the article with DOIs (where available).

If you wish to submit your data to Dryad (<http://datadryad.org/>) and have not already done so you can submit your data via this link [http://datadryad.org/submit?journalID=RSPB&manu=\(Document not available\)](http://datadryad.org/submit?journalID=RSPB&manu=(Document%20not%20available)), which will take you to your unique entry in the Dryad repository.

Online supplementary material will also carry the title and description provided during submission, so please ensure these are accurate and informative. Note that the Royal Society will not edit or typeset supplementary material and it will be hosted as provided. Please ensure that

the supplementary material includes the paper details (authors, title, journal name, article DOI). Your article DOI will be 10.1098/rspb.[paper ID in form xxxx.xxxx e.g. 10.1098/rspb.2016.0049].

Please submit a copy of your revised paper within three weeks. If we do not hear from you within this time your manuscript will be rejected. If you are unable to meet this deadline please let us know as soon as possible, as we may be able to grant a short extension.

Best wishes,
Dr Daniel Costa
mailto: proceedingsb@royalsociety.org

Associate Editor
Board Member: 1
Comments to Author:

While both reviewers consider the manuscript positively, they have identified areas in which the authors could improve the manuscript. I have also gone through the manuscript and identified areas in which the text could be clarified to improve readability and understanding of content and in particular definition of provinces. Please consider the helpful suggestions provided by the reviewers in revising the manuscript.

Reviewer(s)' Comments to Author:

Referee: 1

Comments to the Author(s)

The authors have provided a robust analytical approach to delineate biogeographical provinces of Australian marine molluscs and validated the results using an independent data set. I enjoyed reading this manuscript and appreciate the approach taken, being using ecological data to identify bioregions. I believe this manuscript is suitable for publication, after some clarifications have been made and some minor issues addressed.

The major aspect which I would like clarified is the potential for bioregions/provinces used in the analyses to be acting as proxies for unmeasured environmental and/or geographical variables/features. Within the text the authors have mentioned "Biogeographic factors can in turn be explained by environmental variables, suggesting that environmental controls on diversity may be indirect" (line 17), in this instance does "indirect" refer to context dependent controls, or that environmental controls are moderated by other processes? I think that the relevance of the study would be greatly improved if the authors could clarify the statements such as this through the manuscript (e.g. line 269), as at times these statements appear to be circular. I did appreciate the insight provided on line 270, with important environmental variables differing among bioregions, I believe expanding on this would clarify the author's interpretation of the results. Similarly, the importance of scale of observations could also be mentioned, i.e. if 0.25° cells were used as opposed to 0.5° cells, would the relative importance of environment vs bioregional provinces be likely to differ? I'm not asking the author's to re-analyse the data at varying scales, but it might be worth noting in the discussion.

Minor comments

- Line 7, consider replacing "Has" with "have"
- Line 31, consider moving the comma to after "however"
- Line 33-35, consider describing a couple of examples in the text
- Line 70, - provide RLS internet address (e.g. line 80)
- Line 71, consider replacing "divers in training" with "trained divers"
- Line 80, are there any reasons depth and chlorophyll were not included to represent the environment?
- Line 83, what was the grid cell size/resolution?

- Line 125, would 'distinctiveness' (or similar) be better than 'visibility'?
- Line 129, consider replacing "implement" with "implemented"
- Line 144, maybe list the six abiotic variables here or in results
- Line 153, consider changing "then sampled" to "then randomly sampled" for clarification
- Line 192, is this supposed to be $p < 0.001$?
- Line 194-210, check table references (i.e. Table 3.1 etc)
- Line 235, possibly expand a little on the difference in the nature of diversity between east and west coasts
- Line 235, Please briefly expand on comparison to global studies, does Australia have depauperate tropical fauna or rich temperate fauna by global standards?
- Line 237, consider replacing "some" with "other"
- Line 245, maybe reword, line 182 refers to sharp boundaries on the east coast
- Line 264, consider replacing 'allows' with 'allowed'
- Line 270, I do like this insight
- Line 274-276, Please clarify
- Line 279, reference 15 only considered fish, not invertebrates

Figure "S3.3" caption needs correcting and supplementary numbers need fixing

Referee: 2

Comments to the Author(s)

Dear Dr. Kerr and Dr. Alroy,

I enjoyed reading the paper and believe that the results are very interesting and have a lot to contribute to the field of biogeography. I have very few suggestions and questions. My main concern is related to the Discussion section, which could be improved exploring why distinct biogeographic provinces could better predict species richness than abiotic variables. Please find below my specific comments,

Sincerely,

Hudson Pinheiro

(I waive anonymity)

Marine diversity patterns in Australia are filtered through biogeography

Abstract

Line 11 – What do you mean by biogeographical controls?

Line 16 – it is not clear what are the biogeographic factors – species richness? Species composition? And what about the sharp boundaries? Did you only find evidences for the gradient?

Line 17 – These factors need to be clarified because the richness gradient is a biogeography pattern. The gradient is not related to environmental data but the biogeographic factors are, which in turn explain the gradient.

Line 19 – could you try to add a hypothesis for the direct effects of the biogeographic gradients on the richness patterns. The way it is now the conclusion is that biogeographic gradients explain richness patterns. It is not very clear what you meant because a richness pattern could be interpreted as the same as a biogeographic gradient.

Introduction

Line 24 – and even more important to the field of Biogeography.

Line 31 "biogeographic boundaries affect species richness patterns" – why? How? Do you think it is related to biogeographic barriers? These boundaries mark changes in endemic species, they usually are related to biogeographic barriers or changes in environmental conditions (e.g. transition of tropical to subtropical waters). I suggest you to develop a bit more about this subject.

Line 33 – "temporal changes" – you mean in an ecological or historical scale?

Line 36-38 – some studies on coral reef fishes have included regions and provinces in the models, accounting to the different evolutionary histories of them – See Parravicini et al 2013 (Ecography doi: 10.1111/j.1600-0587.2013.00291.x), Quimbayo et al 2019 (Ecography doi: 10.1111/ecog.03506).
 Line 52-53 – It still not clear what do you mean as biogeographic gradient driving richness patterns because both could be used as synonyms depending on the context. You should describe more straightforward what you mean by biogeographic gradient and how it differ from richness gradient in the Introduction.

General remark - what about the diversity gradients explained by the hypotheses of Center of origin, accumulation, overlap? Mora et al 2003 (10.1038/nature01421.1.) found evidence for the effects of dispersal driving the gradient. Many other studies have found coral reef area as important driver for the diversity gradient as well.

Methods

Line 79 – What about shelf area or coral reef area, which is one of the main factors explaining the diversity of reef organisms, and it is such an important habitat in Australia? Did you try adding them into the model? And what about depth?

Line 155 – “species presences are only controlled by temperature and not directly by geography” – How did you control for the cases where species are restricted to shore or offshore environments? Estuarine dependents or highly mobile species that approach offshore reefs? What if these different geographically habitats have same temperature?

General remarks: what about the description of the general linear models of marine mollusc diversity against abiotic variables?

Results

Figures – All the panels of the Fig 1 are very similar, so I suggest you to leave only one and send the others to supplementary material. In Figure 2, did you try to create a single figure with all the factors together? I think it would work fine and it will reduce the space for the journal.

Line 190 – suggest improving the heading, it is not informative.

Discussion

What I understood is that biogeographic factors is the biogeography of marine organisms in Australia, i.e., subprovinces based on the composition of the species. If the biogeography is more important to predict the richness in the region, the authors should discuss about that. If species composition is more important than the environment, the authors should discuss the great body of work that explore species interactions as determinant to the expected richness found in a province. Seminal papers by Ricklefs are very important to be mentioned. Others by Mora and Bellwood also explore the effects of dispersal and coral species richness and area on the fish richness. The authors could mention the existence of biogeographic barriers as filters, and the role of endemism in determining biogeographic and richness patterns.

Author's Response to Decision Letter for (RSPB-2021-1534.R0)

See Appendix A.

Decision letter (RSPB-2021-1534.R1)

14-Oct-2021

Dear Dr Kerr

I am pleased to inform you that your Review manuscript RSPB-2021-1534.R1 entitled "Marine diversity patterns in Australia are filtered through biogeography" has been accepted for publication in Proceedings B.

The referee(s) do not recommend any further changes. Therefore, please proof-read your manuscript carefully and upload your final files for publication. Because the schedule for publication is very tight, it is a condition of publication that you submit the revised version of your manuscript within 7 days. If you do not think you will be able to meet this date please let me know immediately.

To upload your manuscript, log into <http://mc.manuscriptcentral.com/prsb> and enter your Author Centre, where you will find your manuscript title listed under "Manuscripts with Decisions." Under "Actions," click on "Create a Revision." Your manuscript number has been appended to denote a revision.

You will be unable to make your revisions on the originally submitted version of the manuscript. Instead, upload a new version through your Author Centre.

- 1) A text file of the manuscript (doc, txt, rtf or tex), including the references, tables (including captions) and figure captions. Please remove any tracked changes from the text before submission. PDF files are not an accepted format for the "Main Document".
- 2) A separate electronic file of each figure (tiff, EPS or print-quality PDF preferred). The format should be produced directly from original creation package, or original software format. Please note that PowerPoint files are not accepted.
- 3) Electronic supplementary material: this should be contained in a separate file from the main text and the file name should contain the author's name and journal name, e.g. `authorname_procb_ESM_figures.pdf`

All supplementary materials accompanying an accepted article will be treated as in their final form. They will be published alongside the paper on the journal website and posted on the online figshare repository. Files on figshare will be made available approximately one week before the accompanying article so that the supplementary material can be attributed a unique DOI. Please see: <https://royalsociety.org/journals/authors/author-guidelines/>

4) Data-Sharing and data citation

It is a condition of publication that data supporting your paper are made available. Data should be made available either in the electronic supplementary material or through an appropriate repository. Details of how to access data should be included in your paper. Please see <https://royalsociety.org/journals/ethics-policies/data-sharing-mining/> for more details.

If you wish to submit your data to Dryad (<http://datadryad.org/>) and have not already done so you can submit your data via this link <http://datadryad.org/submit?journalID=RSPB&manu=RSPB-2021-1534.R1> which will take you to your unique entry in the Dryad repository.

Once again, thank you for submitting your manuscript to Proceedings B and I look forward to receiving your final version. If you have any questions at all, please do not hesitate to get in touch.

Sincerely,
Dr Daniel Costa
Editor, Proceedings B
<mailto:proceedingsb@royalsociety.org>

Associate Editor

Comments to Author:

The authors have responded appropriately to all reviewer comments. I have gone through the text and suggested a number of additional editorial corrections associated with wording, clarification of text and standardising nomenclature used that the authors will need to address prior to progressing to publication (attached).

Decision letter (RSPB-2021-1534.R2)

19-Oct-2021

Dear Dr Kerr

I am pleased to inform you that your manuscript entitled "Marine diversity patterns in Australia are filtered through biogeography" has been accepted for publication in Proceedings B.

Your article has been estimated as being 8 pages long. Our Production Office will be able to confirm the exact length at proof stage.

Data Accessibility section

Open Access

Paper charges

Sincerely,

Appendix A

Title: Marine diversity patterns in Australia are filtered through biogeography

Authors: Matthew R Kerr, John Alroy

Response to Reviewers

Reviewer One

The major aspect which I would like clarified is the potential for bioregions/provinces used in the analyses to be acting as proxies for unmeasured environmental and/or geographical variables/features. Within the text the authors have mentioned “Biogeographic factors can in turn be explained by environmental variables, suggesting that environmental controls on diversity may be indirect” (line 17), in this instance does “indirect” refer to context dependent controls, or that environmental controls are moderated by other processes? I think that the relevance of the study would be greatly improved if the authors could clarify the statements such as this through the manuscript (e.g. line 269), as at times these statements appear to be circular. I did appreciate the insight provided on line 270, with important environmental variables differing among bioregions, I believe expanding on this would clarify the author’s interpretation of the results.

- This is an important comment and impelled us to significantly improve the manuscript. We thank the reviewer for these insights. To address these concerns, we added several clarifying statements throughout, particularly in the discussion. These explain the mechanisms driving differences in diversity between bioregions.

Similarly, the importance of scale of observations could also be mentioned, i.e. if 0.25° cells were used as opposed to 0.5° cells, would the relative importance of environment vs bioregional provinces be likely to differ? I’m not asking the author’s to re-analyse the data at varying scales, but it might be worth noting in the discussion.

- This is a good point, and an important one in the literature. We refrained from analysing at different scales because the coarse resolution of the gridded abiotic data would potentially cause issues. To address this comment, we added an explanation and noted at the end of the discussion that higher resolution data would allow for an interesting test of our methods.

- Line 7, consider replacing “Has” with “have”

- This was corrected.

- Line 31, consider moving the comma to after “however”

- This was corrected.

- Line 33-35, consider describing a couple of examples in the text

- Line 70, - provide RLS internet address (e.g. line 80)

- These were added.

- Line 71, consider replacing “divers in training” with “trained divers”

- This was corrected.

- Line 80, are there any reasons depth and chlorophyll were not included to represent the environment?

- These variables are strong predictors in some other studies of marine taxa, and they are important for understanding the gradient. However, this study aimed to use the key abiotic variables featuring in many other marine diversity studies. To be included, high quality data for these variables needed to be available for the whole continent. Thus, chlorophyll had to be left out of this study. We clarified our reasons for choosing variables.
- Depth is additionally a strong correlate, particularly in the open ocean, as this study notes. Due to the spatial scale of this study, and to the fact that the majority of the organisms are coastal and intertidal,

depth (as well as cell area/shelf area) would not be appropriate to consider. We added clarification of this point, stating that a finer-scale study on a more intensely collected taxon including these variables would be an important next step in understanding the relationships between our proposed method and other explanations for marine diversity gradients.

- Line 83, *what was the grid cell size/resolution?*

- This was added.

- Line 125, *would 'distinctiveness' (or similar) be better than 'visibility'?*

- We agree with this suggestion and have made the correction.

- Line 129, *consider replacing "implement" with "implemented"*

- This was corrected.

- Line 144, *maybe list the six abiotic variables here or in results*

- This list was added, and a correction added to include "latitude" in that list of variables.

- Line 153, *consider changing "then sampled" to "then randomly sampled" for clarification*

- This was corrected.

- Line 192, *is this supposed to be $p < 0.001$?*

- This was corrected.

- Line 194-210, *check table references (i.e. Table 3.1 etc)*

- This was an oversight, and the references have been checked and corrected throughout.

- Line 235, *possibly expand a little on the difference in the nature of diversity between east and west coasts*

- This was added.

- Line 235, *Please briefly expand on comparison to global studies, does Australia have depauperate tropical fauna or rich temperate fauna by global standards?*

- We added a sentence speculating on the differences, but we highlight that many studies including Australia only report raw richness and thus cannot be compared to our results here.

- Line 237, *consider replacing "some" with "other"*

- This was corrected.

- Line 245, *maybe reword, line 182 refers to sharp boundaries on the east coast*

- This was corrected.

- Line 264, *consider replacing 'allows' with 'allowed'*

- This was corrected.

- Line 274-276, *Please clarify*

- We have added and moved several sentences in the discussion to clarify our interpretations, in line with this comment and with the initial comments made in the review. In particular, we put forth a clearer mechanism that explains the results and highlighted it throughout the manuscript.

- Line 279, *reference 15 only considered fish, not invertebrates*

- This line was added to highlight the previous use of the dataset and was unclear. It was removed to instead focus the paragraph on the results of this study.

Figure "S3.3" caption needs correcting and supplementary numbers need fixing

- This was an oversight: all supplementary figures have been changed to be in the correct format.

Reviewer Two

Line 11 – What do you mean by biogeographical controls?

- We rephrased this sentence in the abstract to make the language clearer.

Line 16 – it is not clear what are the biogeographic factors – species richness? Species composition? And what about the sharp boundaries? Did you only find evidences for the gradient?

- We rephrased this point to highlight that the biogeographic gradients are based on species composition and are fundamentally independent of diversity.

Line 17 – These factors need to be clarified because the richness gradient is a biogeography pattern. The gradient is not related to environmental data but the biogeographic factors are, which in turn explain the gradient.

- This has been clarified.

Line 19 – could you try to add a hypothesis for the direct effects of the biogeographic gradients on the richness patterns. The way it is now the conclusion is that biogeographic gradients explain richness patterns. It is not very clear what you meant because a richness pattern could be interpreted as the same as a biogeographic gradient.

- This has been clarified, and a mechanism has now been proposed here and throughout the manuscript.

Line 24 – and even more important to the field of Biogeography.

- We agree, and we have added this to highlight that this study combines macroecology and biogeography.

Line 31 “biogeographic boundaries affect species richness patterns” – why? How? Do you think it is related to biogeographic barriers? These boundaries mark changes in endemic species, they usually are related to biogeographic barriers or changes in environmental conditions (e.g. transition of tropical to subtropical waters). I suggest you to develop a bit more about this subject.

- This is an important section of the introduction and was mentioned by both reviewers and throughout this review. To address this comment, we included a description of the mechanism and explained how it relates to definitions of biogeography. We have also highlighted this point elsewhere in this response, making appropriate changes to clarify it.

Line 33 – “temporal changes” – you mean in an ecological or historical scale?

- This was corrected to reflect the historical and palaeontological time scale of the reference.

Line 36-38 – some studies on coral reef fishes have included regions and provinces in the models, accounting to the different evolutionary histories of them – See Parravicini et al 2013 (Ecography doi: 10.1111/j.1600-0587.2013.00291.x), Quimbayo et al 2019 (Ecography doi: 10.1111/ecog.03506).

We thank the author for highlighting these important papers, and we have included citations of them both in the introduction and elsewhere. In particular, the Parravicini et al. paper is very relevant, and it enhances the points made in this manuscript.

Line 52-53 – It still not clear what do you mean as biogeographic gradient driving richness patterns because both could be used as synonyms depending on the context. You should describe more straightforward what you mean by biogeographic gradient and how it differ from richness gradient in the Introduction.

- We clarified the language used here and elsewhere in the manuscript.

General remark - what about the diversity gradients explained by the hypotheses of Center of origin, accumulation, overlap? Mora et al 2003 (10.1038/nature01421.1.) found evidence for the effects of dispersal driving the gradient. Many other studies have found coral reef area as important driver for the diversity gradient as well.

- These hypotheses are interesting, and many studies compare them to environmental explanations. Our manuscript, methods, and data have only been used here to test biogeographical approaches. Although these hypotheses are out of scope, a follow-up study concerning them would potentially be very interesting. We have added comments addressing this point to the manuscript, primarily in the discussion.

Methods

Line 79 – What about shelf area or coral reef area, which is one of the main factors explaining the diversity of reef organisms, and it is such an important habitat in Australia? Did you try adding them into the model? And what about depth?

- These are important covariates in many studies. Due to the nature of the data we used, which included equal-area coastal cells, they could not be included in our models. We added a sentence stating this more clearly. Studies on the secondary RLS dataset have included depth, but the depth range is extremely narrow. We also clarify this point and suggest that a future study that could use higher resolution data.

Line 155 – “species presences are only controlled by temperature and not directly by geography” – How did you control for the cases where species are restricted to shore or offshore environments? Estuarine dependents or highly mobile species that approach offshore reefs? What if these different geographically habitats have same temperature?

- This was a key point of the simulation, so we have clarified how the simulation works. In addition to not restricting species by geographical distribution, we removed habitat and dispersal requirements. This allowed us to test if environmental variables alone could recreate the patterns we observe or if the biogeographic divisions are required for the maintenance of the gradient.

General remarks: what about the description of the general linear models of marine mollusc diversity against abiotic variables?

- We included an in-text description of these models. Although many studies include a full model description, we do not feel that writing out the model formula is appropriate in this case.

All the panels of the Fig 1 are very similar, so I suggest you to leave only one and send the others to supplementary material..

- We carefully considered this comment, but we believe that the differences between the methods are a key result in the paper. Many current studies only focus on one diversity measure (often raw richness) and the inclusion of four in this manuscript is a repeated theme (Table 1, 2). Although the figure is large and the panels appear to be similar, we have left the figure in the same place as in the original submission to highlight these crucial differences.

In Figure 2, did you try to create a single figure with all the factors together? I think it would work fine and it will reduce the space for the journal

- In an earlier version of the manuscript we included all provinces on one figure, but we found that the visualisation of blending between the regions was difficult. To address this comment, we included a single all-factor figure in the supplement.

Line 190 – suggest improving the heading, it is not informative.

- We changed the heading to highlight which analysis was being discussed.

What I understood is that biogeographic factors is the biogeography of marine organisms in Australia, i.e., subprovinces based on the composition of the species. If the biogeography is more important to predict the richness in the region, the authors should discuss about that. If species composition is more important than the environment, the authors should discuss the great body of work that explore species interactions as determinant to the expected richness found in a province. Seminal papers by Ricklefs are very important to be mentioned. Others by Mora and Bellwood also explore the effects of dispersal and coral species richness and

area on the fish richness. The authors could mention the existence of biogeographic barriers as filters, and the role of endemism in determining biogeographic and richness patterns.

- We thank the reviewer very much for these comments. We made several changes towards the end of the discussion to address this matter. In particular, we more clearly highlight how our results and proposed mechanism fit in with the above-mentioned work, both in the manuscript and in this review. We added citations of these papers and expanded the discussion around them.